# Singular Value Adaptation for Parameter-Efficient Fine Tuning

## Abstract

Parameter-Efficient Fine-Tuning (PEFT) has become a crucial approach in handling the growing complexity of large models and vast datasets across multiple fields such as Computer Vision or Natural Language Processing. Among the most promising of these methods are Low-Rank Adaptation (LoRA) and its derivatives, which fine-tune a pre-trained weight matrix $\mathbf{W}$ by introducing a low-rank update matrix $\mathbf{\Delta W}$. While these approaches have demonstrated strong empirical performance, they remain largely heuristic, with little theoretical grounding to explain their behavior or guide the design of $\mathbf{\Delta W}$ for different objectives. This lack of theoretical insight limits our understanding of when these methods are most effective and how they can be systematically improved. In this paper, we propose a theoretical framework for analyzing and designing LoRA-based methods, with a focus on the formulation of $\mathbf{\Delta W}$. By establishing a deeper understanding of the interplay between $\mathbf{W}$ and $\mathbf{\Delta W}$, we aim to enable more efficient and targeted fine-tuning strategies, opening the door to novel variants that strike an optimal balance between performance and efficiency. Our proposed method - **Si**ngular **V**alue **A**daptation - uses insights from our theoretical framework to incorporate inductive biases on the formulation of $\mathbf{\Delta W}$, leading to a PEFT method that is up to $50\times$ more parameter efficient that LoRA, while achieving comparable or better performance across various vision and language tasks.

## 1 Introduction

Large pre-trained neural networks have become indispensable across a wide array of domains, including natural language processing and computer vision, but fine-tuning them efficiently for specialized downstream tasks remains a significant challenge. The sheer scale of modern models makes fine-tuning all parameters computationally expensive and often impractical. To address this, various Parameter-Efficient Fine-Tuning (PEFT) methods have been developed, aiming to optimize model adaptation while minimizing computational overhead. PEFT techniques, such as Adapters, Prompts and Prefixes, Low-Rank Adaptation (LoRA), and their numerous variants, have gained popularity due to their ability to fine-tune models by only modifying a small number of parameters, making the process more efficient. These methods have been successfully deployed across various applications, demonstrating impressive performance gains while significantly reducing resource requirements. LoRA and its derivatives have in particular become widely prevelant, owing to the fact that these methods give comparable or improved performance over other PEFT methods, and do not incur any additional computational costs over the base model during inference.

Despite their success, there remains a lack of clarity regarding the underlying mechanisms that make LoRA and its derivatives effective. Most formulations of the low-rank update matrix are largely heuristic, with no studies on how different formulations affect the final merged matrix at inference. Our analysis on the original pretrained and and adapted weight matrices of a ViT-Base model finds that they do not significantly differ in terms of their rank (see table 1). Our findings reveal that most pre-trained matrices are in fact full-rank or near-full-rank, all. Instead, in this paper, we hypothesize that the effectiveness of LoRA based techniques may be driven by changes in the "effective rank" (Roy & Vetterli, 2007) of the model's weight matrices during fine-tuning. In table 1, we show initial observations supporting this hypothesis, where increases in the effective rank correlate with improved model performance, as evidenced by our sample results (Fig 1 (left)).

Building on this observation, we propose a novel method explicitly designed to maximize the increase in effective rank under the constraint of few trainable parameters.

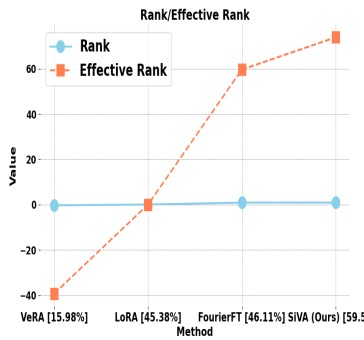 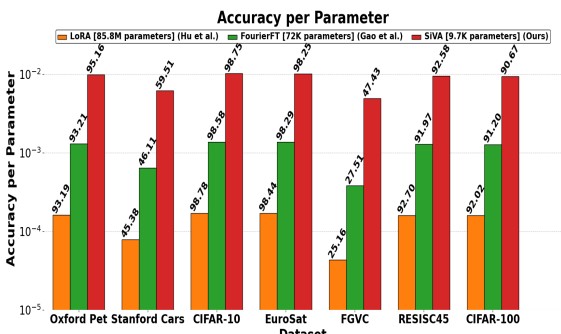

Figure 1: *(Left)* Plot of change in rank and effective rank of various PEFT methods, relative to an average baseline rank of 766.92 and an average baseline effective rank of 520.03 obtained from pre-trained ViT-Base model Query and Value matrices. Accuracy over Base model is computed using a linear probe (LP). The $x$-axis lists method names along with their respective accuracies on Stanford Cars dataset. *(Right)* Plot of accuracy per parameter for various PEFT methods, based on experiments conducted across 7 datasets using ViT-Base architecture. Accuracy for each method is displayed above the corresponding bar, number of trainable parameters for each method is provided in legend. Our PEFT method, SiVA, achieves significant parameter reduction while maintaining high accuracy, consistently delivering best accuracy per parameter performance.

Our method, **Si**ngular **V**alue **A**daptation (**SiVA**), is grounded in theoretical insights about the relationships between the structure of the update matrix, its constraints on the number of trainable parameters, and its impact on the effective rank on the final matrix post training. By focusing on increasing the effective rank while maintaining a minimal parameter footprint, **SiVA** leverages the strengths of low-rank adaptation techniques while introducing new mechanisms to optimize effective rank growth. The contribution to the final performance by each trained

| Method | Accuracy | Parameters | Rank | Effective Rank |
|---|---|---|---|---|
| Base + LP | 25.76 | - | 766.92 | 520.03 |
| LoRA | 45.38 | 581K | 766.97 (+0.05) | 520.06 (+0.03) |
| FourierFT | 46.11 | 72K | 767.85 (+0.90) | 579.61 (+59.6) |
| VeRA | 15.98 | 48K | 767.26 (+0.34) | 480.84 (-39.2) |
| SiVA (Ours) | 59.51 | 9.7K | 767.85 (+0.93) | 593.86 (+73.8) |

Table 1: Table depicting the accuracy over the classification task and the effective rank of the learned adapters on the Stanford Cars dataset over the ViT-B16 model. This illustrates that while the rank of the adapters are approximately equal across all methods, the accuracy seems to significantly correlate with its effective rank. Our method has the highest effective rank amongst the SoTA methods presented above as well as an accuracy score $\approx 13.5\%$ above them. The values in the brackets indicate the increase in rank/effective rank over the base model.

parameter of SiVA - measured as *performance per parameter (PPP)* - is significantly higher compared to existing methods, even those that aggressively attempt to minimize the parameters via heuristic formulations of the LoRA update matrix (Kopiczko et al., 2024; Gao et al., 2024). A visualization of the PPP for different methods on an image classification task is shown in Fig 1(right), with additional visualizations shown later with other experimental results.

(1) The first component of our overarching theory makes use of the observation that the effective rank of a matrix is solely a function of its singular values. Based on this insight, we derive a relation for the contribution made by each individual singular value update to the effective rank of the merged matrix. (2) Next, we realize that to achieve parameter efficiency, we only need to update (train) a subset of all singular values in each weight matrix of the base model. This leads us to derive that the smaller singular values play a greater role in maximizing the effective rank of the final matrix, and thus they are the ones that should be tuned, while the larger singular values can be left unchanged. (3) Finally, we prove the intuitive result that the left and right singular vectors of the adapted matrix should be aligned with those of the pre-trained weight matrix to maximize the effective rank (of the adapted matrix). Overall, these three components reveal a simple formulation of the low-rank update matrix - the update matrix can be written as the composition of three matrices $\mathbf{U}$, $\mathbf{V}$, and $\mathbf{S_{\Delta W}}$, as $\mathbf{\Delta W} = \mathbf{U S_{\Delta W} V^T}$, where $\mathbf{U}$ and $\mathbf{V}$ are the left and right singular vectors of a pre-trained weight matrix, the lower elements of the diagonal matrix $\mathbf{S_{\Delta W}}$ are learned using Gradient Descent, and the other elements of $\mathbf{S_{\Delta W}}$ are set to zero. (4) To conclude our theoretical insights, we show that this formulation can achieve the minimum with zero loss in the least squares regression problem when all singular values are trainable.

Our overall contributions can be summarized as follows: (i) We show that the performance of transformer models on downstream tasks is correlated to the effective ranks of the query and value matri-

ces, an insight that was previously unknown; (ii) We propose a specialized structure for the composition of the update matrix, and theoretically show that the proposed structure maximizes the effective rank of the merged weights post-training. We propose a simple, yet highly performant LoRA-based approach inspired by our theoretical insights; (iii) We evaluate the performance of the proposed approach across several tasks spanning both Computer Vision and Natural Language Processing, and show that our approach, despite having a simple formulation, outperforms state-of-the-art methods in terms of performance-per-parameter by large margins. In terms of absolute performance across different metrics, our approach achieves results comparable to or better than existing PEFT methods.

## 2 RELATED WORK

**LoRA and Variants.** LoRA-based PEFT methods solve the computational overhead of fine-tuning large models by modelling the change in parameters as a low-rank matrix. Various decompositions of the low-rank matrix have been proposed, leading to an entire family of LoRA derivatives (Kopiczko et al., 2024; Liu et al., 2024; Gao et al., 2024; Aghajanyan et al., 2021; Karimi Mahabadi et al., 2021; Edalati et al.; Liao et al., 2023; He et al., 2023). The original LoRA formulation (Hu et al., 2022) decomposed all weight matrices as a product of two rectangular learnable matrices of specified rank. Dynamic-rank LoRA derivatives (Zhang et al., 2023; 2024; Ding et al., 2023; Valipour et al., 2023; Haobo et al., 2024) methods further refine this approach by using adaptive ranks for different layers. More recent works propose the use of pseudo-random vectors or matrices to achieve aggressive parameter compression. NOLA (Koohpayegani et al., 2024) learns the coefficient of linear combination of pseudo-random matrices, while FourierFT (Gao et al., 2024) samples a random spectral basis and learns the sparse spectral coefficients , using an Inverse Discrete Fourier Transform to get the weight update matrix. In similar spirit, VeRA (Kopiczko et al., 2024) samples two random matrices, and scales them by learnable factors before multiplying them to obtain the update matrix. These methods are largely heuristic, without any underlying common principle that guides the formulation of the weight update matrix.

**Other PEFT Methods.** Adapter Tuning methods (Pfeiffer et al., 2021; Houlsby et al., 2019b; Lei et al., 2023; He et al.; Zhu et al., 2021) introduce task-specific parameters through small layers (adapters) inserted within a pre-trained model. Prompt/Prefix Tuning methods (Li et al., 2023; Liu et al.; Zhang et al.; Zhu & Tan; Wu et al., 2022; Ma et al., 2022; Lester & Constant; Liu et al., 2023) prepend learnable vectors to the inputs of the model (prompts) or individual intermediate layers (prefixes). Some other methods include BitFit (Zaken et al., 2022) which only tunes the model biases and IA3 (Liu et al., 2022) which introduces additional parameters in the Self-Attention module. All these methods increase the complexity of the base model, leading to increased inference times and a requirement for additional space. LoRA derivatives do not suffer from this issue as the newly learned parameters can be directly added to the parameters of the base model post training.

## 3 SIVA: FORMULATION AND METHODOLOGY

### 3.1 NOTATIONS AND PRELIMINARIES

Our work is based on the hypothesis that the performance of the adapted model is proportional to the increase in effective rank caused by adding the learned matrix $\mathbf{\Delta W}$ to the pre-trained weight matrix $\mathbf{W}$. Crucially, we want to maximize the effective rank while keeping the number of parameters in $\mathbf{\Delta W}$ minimum. Since adapter-based methods are usually applied to the square-shaped query and value matrices in transformers, we restrict our formulation to square matrices. Formally, the **effective rank** (Roy & Vetterli, 2007) of a square matrix $\mathbf{A} \in \mathbb{R}^{n \times n}$ is given by

$$\texttt{erank}(\mathbf{A}) = e^{\mathcal{H}_{\mathbf{A}}} = e^{\mathcal{H}_{\mathbf{\Sigma_A}}},$$

where $\Sigma_{\mathbf{A}}$ represents the diagonal matrix of singular values $(\sigma_1, \sigma_2, ..., \sigma_n)$ of $\mathbf{A}$. Here $\mathcal{H}_{\mathbf{\Sigma_A}} := -\sum_{i=1}^{n} p_i log(p_i)$, where $p_i$ is the $i^{\text{th}}$ normalized singular value computed as $p_i = \frac{\sigma_i}{\sum_{j=1}^{n} \sigma_k}$. We refer to $\mathcal{H}_{\mathbf{A}}$ as the *entropy* of matrix $\mathbf{A}$. Furthermore, for a discrete random variable $\chi$ that obeys a distribution $\mathcal{P}$ with finite support over $n$ states $(\bigcup_{i=1}^{n} \chi_i)$, we define $H(\mathcal{P}) = -\sum_{i=1}^{n} p_i \log(p_i)$, where $p_i = \texttt{Prob}(\chi = \chi_i)$. In this work, we restrict ourselves to dealing with discrete random variables alone.

We posit that after fine-tuning the adapters attached to the query and value matrices, the effective rank of the combined matrices should be greater than the original matrix to increase performance on the downstream task. In other words, if $\mathbf{W}$ represents the original weight matrix of an adapted query

or value, and $\mathbf{\Delta W}$ represents the weight matrix provided by the adapter module *post fine-tuning on a downstream task*, then $\texttt{erank}(\mathbf{W} + \mathbf{\Delta W}) > \texttt{erank}(\mathbf{W})$. We observe that the effective rank of the resulting matrix depends solely on its singular values. Based on this observation, we now present our theory to analyze the conditions under which $\mathbf{W} + \mathbf{\Delta W}$ has the highest effective rank over $\mathbf{W}$ under certain constraints on $\mathbf{\Delta W}$.

## 3.2 THEORETICAL FORMULATION

Since the effective rank increases monotonically with the entropy $\mathcal{H}$, we shift our focus towards the analysis of $\mathcal{H}$. We first derive an expression for computing the change in entropy of the combined matrix over the original matrix, i.e. $\Delta\mathcal{H} = \mathcal{H}_{\mathbf{W}+\mathbf{\Delta W}} - \mathcal{H}_{\mathbf{W}}$. Using this expression, we proceed to show that for maximizing $\Delta\mathcal{H}$, the change in the $i$th singular value of $\mathbf{W}$ should be proportional to its own magnitude.

The next part of our analysis uses this observation to find the ideal singular values to learn given that we wish to constrain the number of parameters (here, singular values) that are trained. We proceed to show that if all singular values are not freely tunable, it is optimal to tune the lower singular values of $\mathbf{W}$ using $\mathbf{\Delta W}$. Additionally, we further show that for two different $\mathbf{\Delta W}$ matrices that have the same norm and the same number of tunable singular values, the one that has its singular vectors aligned with the singular values of $\mathbf{W}$ results in the maximum increase in the effective rank.

We then show that this formulation - training only the lower singular values of $\mathbf{W}$ by adding an appropriately crafted $\mathbf{\Delta W}$ - achieves an optimal solution for the Linear Regression Problem, and the proceed to adapt this method for fine-tuning transformers.

**Lemma 1** (Change in Entropy). *Given a random variable $\chi$ and any two distributions $\mathcal{P}$, $\mathcal{Q}$ over $(\bigcup_{i=1}^{n} \chi_i)$, define $\delta p_i := \mathcal{P}(\chi = \chi_i) - \mathcal{Q}(\chi = \chi_i)$, and assume that such that $|\delta p_i| << 1$, $\forall 1 \leq i \leq n$. Then the change in entropy $\Delta H_{q,p} := H(q) - H(p)$ is given by: $\Delta H_{q,p} = -\sum_{i=1}^{n} \delta p_i (\log p_i)$.*

Our primary result investigates the optimal perturbations introduced by the adapter, $\mathbf{\Delta W}$, to the pre-trained weight matrix, $\mathbf{W}$. The proof outline begins by deriving a simplified representation of the singular values of the adapted weight matrix, $\mathbf{W} + \mathbf{\Delta W}$, utilizing a first-order Taylor expansion around the normalized singular values. A first-order approximation is sufficient in this case, as the adjustment to the $i^{\text{th}}$ singular value, $\delta\sigma_i$, is generally much smaller in magnitude compared to the corresponding singular value of $\mathbf{W}$.

Subsequently, we employ our central technical Lemma 1 to analyze the distributions induced by the normalized singular values of $\mathbf{W} + \mathbf{\Delta W}$ and $\mathbf{W}$, respectively. Maximizing the relevant entropy reduces the problem to optimizing an objective function over the free variables $\delta\sigma_i$. To account for regularization constraints on $\mathbf{\Delta W}$, a common practice in the PEFT literature (Hu et al., 2022), we incorporate a constraint on the Frobenius norm of $\mathbf{\Delta W}$. The optimal adjustments for this constrained optimization are then derived using the method of Lagrange multipliers. The question of whether these optimal adjustments scale linearly with the magnitude of the corresponding singular values in $\mathbf{W}$ is resolved affirmatively in the following theorem.

**Theorem 1** (Increase in Entropy under Singular Value Adjustment). *Let $A \in \mathbb{R}^{n \times n}$ be a matrix with singular values $\sigma_i^A \geq 0$ for $i = 1, 2, \ldots, n$, and let $S_A = \sum_{i=1}^{n} \sigma_i^A$. Consider perturbations $\delta\sigma_i \in \mathbb{R}$ such that the singular values of $A + B$ become $\sigma_i^{A+B} = \sigma_i^A + \delta\sigma_i$, for an arbitrary matrix $B \in \mathbb{R}^{n \times n}$. Under the constraint $\|B\|_F^2 = \sum_{i=1}^{n} (\delta\sigma_i)^2 \leq C$, the maximum possible change in entropy $\Delta H$ is:*

$$\Delta H_{\max} = \sqrt{C} \cdot \sqrt{\sum_{i=1}^{n} c_i^2}, \tag{1}$$

*where $c_i = -\dfrac{\left(\log\left(\frac{\sigma_i^A}{S_A}\right) + \mathcal{H}_A\right)}{S_A}$. The optimal adjustments $\delta\sigma_i$ are given by:*

$$\delta\sigma_i = c_i \cdot \frac{\sqrt{C}}{\sqrt{\sum_{j=1}^{n} c_j^2}}. \tag{2}$$

All proofs are provided in the Appendix. Even if one optimizes the singular values of $\mathbf{W} + \mathbf{\Delta W}$ to follow the structure outlined in Theorem 1, a critical question remains: how does this optimization

contribute to parameter reduction when one might need to train $n$ such singular values? This could lead to the training of $\Delta \mathbf{W} \in \mathbb{R}^{n \times n}$, resulting in as many parameters as the original pre-trained matrix, making it inefficient from a parameter-efficiency standpoint.

To address this, we introduce Theorem 2. It shows that if updates are restricted to only some $k$ of the singular values of $\mathbf{W}$, while the remaining $n-k$ singular values remain unchanged, it is optimal under mild conditions to update the $k$ smallest singular values. This can be intuitively expected in the following sense: the entropy of a discrete random variable is maximized when the probabilities of it taking different values are equi-probable. Formally, this result is derived by tweaking Equation 1 and utilizing the properties of the constants $c_i$, as detailed in Theorem 1, which are further explored in the Appendix.

**Theorem 2** (Sparse, Optimal Modification of Singular Values to Maximize Entropy Increase). *Let $A, B \in \mathbb{R}^{n \times n}$ be two matrices, with $A$ being fixed. Suppose we are allowed to perturb at most $k$ singular values of $A+B$ (i.e., at most $k$ of the $\delta\sigma_i$ are non-zero), using a matrix $B \in \mathbb{R}^{n \times n}$ under the constraint $\|B\|_F^2 = \sum_{j=1}^{k} (\sigma_j^B)^2 \leq C$. To maximize the increase in entropy $\Delta H = \mathcal{H}_{A+B} - \mathcal{H}_A$, it is optimal to modify the $k$-smallest singular values of $A$.*

Theorem 2 offers a significant reduction in the number of parameters that need to be trained, depending on the desired downstream task performance (these requirements are encoded in the values of $k$ and $C$). However, this still involves learning parameters for the orthonormal matrices $U_{\Delta \mathbf{W}}$ and $V_{\Delta \mathbf{W}}$, which are part of the singular value decomposition (SVD) of $\Delta \mathbf{W}$. To further reduce the number of parameters, we focus on minimizing the parameters to be trained in these orthonormal matrices. It is easy to observe that no additional parameters would be required to learn these matrices if, after training, the singular vectors of $\Delta \mathbf{W}$ align with those of $\mathbf{W}$, irrespective of the initialization of the entries of $\Delta \mathbf{W}$. In fact, under mild conditions, this is precisely what we prove in Theorem 3. This result is essential in justifying our formulation, where we explicitly constrain the orthonormal matrices in the SVD of $\Delta \mathbf{W}$ to be identical to those of $\mathbf{W}$. This provides the second major reduction in parameters compared to other PEFT methods, as we ultimately only train scalar values representing adjustments to the pre-trained weights $\mathbf{W}$. The proof primarily follows a greedy argument, relying on Lemma 1 and a result from first-order perturbation theory (Stewart, 1998).

Learning coefficients for linear combinations has been explored in prior work (Koohpayegani et al., 2024; Kopiczko et al., 2024; Gao et al., 2024), where adapter weights are formulated as linear combinations of matrices from a basis of randomly chosen matrices. In contrast, our approach utilizes fixed deterministic matrices derived from the pre-trained model. An additional advantage of our method is that it remains invariant to the implementation of random number generators, unlike these other methods, and it does not require storing random seeds post-training. Furthermore, our approach removes a significant constraint required by previous methods: we do not require the adapter matrices to span the same random basis at every layer. Instead, at each layer the singular vectors of the adapter matrix are aligned with the singular vectors of the corresponding weight matrix, enhancing the expressivity of our method while avoiding cross-talk between weight matrices across layers.

**Theorem 3** (Alignment of Singular Vectors w.r.t Pretrained Weights). *Let $A \in \mathbb{R}^{n \times n}$ be a matrix with singular values $\sigma_i^A$ arranged in descending order ($\sigma_1^A \geq \sigma_2^A \geq \cdots \geq \sigma_n^A \geq 0$) and corresponding left and right singular vectors $\mathbf{u}_i^A$ and $\mathbf{v}_i^A$. Let $B \in \mathbb{R}^{n \times n}$ be a fixed matrix with exactly $k$ nonzero singular values $\sigma_j^B$ (with $\sigma_1^B \geq \sigma_2^B \geq \cdots \geq \sigma_k^B > 0$) and corresponding singular vectors $\mathbf{u}_j^B$ and $\mathbf{v}_j^B$. Under the constraint $\|B\|_F^2 = \sum_{j=1}^{k} (\sigma_j^B)^2 = C$, the maximum increase in entropy $\Delta H = \mathcal{H}_{A+B} - \mathcal{H}_A$ is achieved when the following happen, in order:*

1. *First, the singular vectors of $B$ corresponding to its largest singular value are aligned with the singular vectors of $A$ corresponding to its smallest singular value; specifically, $\mathbf{u}_j^B = \mathbf{u}_{n-j+1}^A$ and $\mathbf{v}_j^B = \mathbf{v}_{n-j+1}^A$ for $j = 1, 2, \ldots, k$.*

2. *Since the largest singular value of $B$ is now aligned, $\Delta H$ is further maximized by aligning the next largest singular value of $B$ with the next smallest singular vector of $A$, and so on recursively, for all $k$ singular values of $B$.*

*Therefore, the largest increase in entropy is achieved by aligning the singular values of $B$ in decreasing order with the singular vectors of $A$ in increasing order of their indices.*

Theorems 1–3 provide a strong theoretical foundation for the PEFT formulation that we introduce as **SiVA**. However, to rigorously establish that **SiVA** achieves optimal performance, we need to connect this formulation directly to the loss function used during training. It is important to note that we present experiments for **SiVA** on large, deep networks, such as ViT-B16 (small/large), which incorporate multiple non-linearities and attention heads, thereby inducing complex inductive biases into the learned weights. Fully characterizing the trajectory of the solutions derived by **SiVA** under these conditions is highly non-trivial and left as future work.

Instead, we show that in a simpler Linear Regression setting, there exist weight matrices $\mathbf{\Delta W}$ whose singular vectors align with those of $\mathbf{W}$ (**SiVA** form) and are within the set of weight matrices that minimize the loss function. The proof uses a standard proof-by-contradiction approach: we begin by assuming that no such solution exists, and then construct an optimal solution of the **SiVA** form for the regression task. By demonstrating the existence of this optimal solution, we invalidate the original assumption, proving that a **SiVA**-form solution is optimal for the linear regression task. The proof is constructive and can be derived using standard derivative-based optimization techniques.

**Theorem 4** (**SiVA** Style Solutions Lie in Set of Minimizers for Linear Regression). *Let $A \in \mathbb{R}^{m \times n}$ be a full-rank matrix with singular value decomposition $A = U_A \Sigma_A V_A^\top$, where $U_A \in \mathbb{R}^{m \times n}$ and $V_A \in \mathbb{R}^{n \times n}$ are orthogonal matrices, and $\Sigma_A \in \mathbb{R}^{n \times n}$ is a diagonal matrix with positive entries $\sigma_{A1}, \sigma_{A2}, \ldots, \sigma_{An}$. Let $x \in \mathbb{R}^n$ and $y \in \mathbb{R}^m$ be given vectors. Define $B = U_A S V_A^\top$, where $S \in \mathbb{R}^{n \times n}$ is a diagonal matrix with entries $\sigma_1, \sigma_2, \ldots, \sigma_n$. Consider the Mean Squared Error:*

$$L(\sigma_1, \ldots, \sigma_n) = \|(A + B)x - y\|^2 .$$

*Then, $L(\sigma_1, \ldots, \sigma_n)$ is minimized by choosing*

$$\sigma_i = \frac{u_i^\top y}{v_i^\top x} - \sigma_{Ai},$$

*for each $i$ such that $v_i^\top x \neq 0$, where $u_i$ and $v_i$ are the $i$-th columns of $U_A$ and $V_A$, respectively.*

### 3.3 METHOD

This section outlines the implementation of our method. We use insights from our theoretical observations to develop a PEFT method that is conceptually simple, aggressively parameter-efficient, and highly performant. Our approach directly draws inspiration from our theoretical framework, and is based on the following hypothesis - *Modification of a small subset of singular values of the pre-trained model weights is sufficient to achieve high performance, while simultaneously limiting the number of parameters to be trained*. We formulate $\mathbf{\Delta W}$ such that it selectively updates the lower singular values of a pre-trained weight matrix $\mathbf{W}$ when added to it. The pseudocode outlining the various components of SiVA is presented in Algorithm 1.

Formally, let $\mathbf{W}$ be the weight matrix to be adapted to a downstream task by adding the update matrix $\mathbf{\Delta W}$. We compose $\mathbf{\Delta W}$ as product of three matrices $\mathbf{U}$, $\mathbf{S}$, and $\mathbf{V^T}$, where $\mathbf{U}$ and $\mathbf{V}$ are the left and right singular vectors of $\mathbf{W}$ respectively (Theorem 4). Note that this initialization of $\mathbf{U}$ and $\mathbf{V}$ is a one-time operation, performed only when attaching the **SiVA** module to the base layer. $\mathbf{S}$ is a diagonal matrix whose values are optimized using Gradient Descent.

To achieve parameter efficiency, we only train $k$ diagonal elements of $\mathbf{S}$, corresponding to the singular values of $\mathbf{W}$ that need to be modified. The remaining singular values are frozen, along with $\mathbf{U}$ and $\mathbf{V^T}$. Only the lowest singular values are chosen for training (Theorem 3). Analogous to the observation made by AdaLoRA (Zhang et al., 2023) about the suboptimal choice of having the same rank for each update matrix, maintaining a constant number of $\mathbf{k}$ per layer can also be suboptimal. Therefore, we determine $\mathbf{k}$ in each layer based on the variance captured by the corresponding eigenvalues. We pick the singular values corresponding to the eigenvalues that capture the bottom $(1 - perc)\%$ of the variance, where $perc$ is a hyperparameter. The pseudocode for computing $k$ for a given value of $perc$ is presented in the *get_values_and_vectors* function in Algorithm 1.

## 4 EXPERIMENTS AND RESULTS

We evaluate the performance of SiVA across computer vision (CV) and natural language processing (NLP) datasets. For CV, SiVA fine-tunes **(1)** vision transformers (ViT) (Dosovitskiy et al., 2021) in Base and Large variants for image classification. For NLP, SiVA is applied to **(2)** RoBERTa-Large (Liu et al., 2020) for natural language understanding on the GLUE (Wang et al., 2018) benchmark,

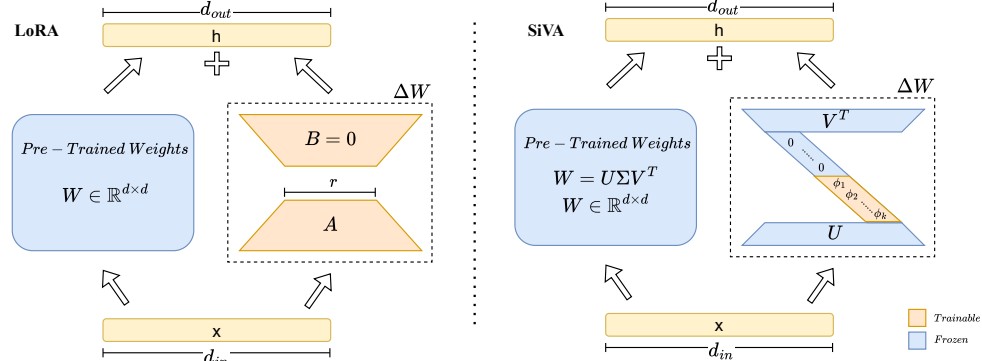

Figure 2: Visualization of LoRA (left) against SiVA (right). Compared to LoRA, SiVA has significantly fewer trainable components. Additionally, all frozen components in SiVA are derived from the base matrix itself.

**Algorithm 2** PyTorch-style Pseudocode for our method, SiVA

```python
class SiVA(nn.Module):
    def __init__(
        self,
        perc: float, # only train the singular values corresponding to the eigenvalues
        # capturing bottom (1 - perc)% variance
        base_layer: nn.Module # pre-trained layer,
    ):
        # definitions
        self.perc = perc
        self.base_layer = base_layer

        # Get the the vector of lower singular values,
        # and corresponding left and right singular vectors
        self.U, S, self.V = self.get_values_and_vectors(base_layer.weight)
        self.S = nn.Parameter(torch.zeros_like(S))

    def get_values_and_vectors(self, matrix):
        U, S, V = torch.svd(matrix)
        # Calculate the total variance (sum of squared singular values)
        total_variance = (S**2).sum()

        # Calculate the cumulative sum of squared singular values
        cumulative_variance = torch.cumsum(S**2, -1)

        # Find the number of singular values that capture perc% of the variance
        num_frozen_singular_values = torch.searchsorted(
            cumulative_variance,
            self.perc * total_variance
        ) + 1

        num_trainable_singular_values = S.shape[0] - num_singular_values

        reduced_U = U[:, -num_trainable_singular_values:]
        reduced_S = S[-num_trainable_singular_values:]
        reduced_V = V[:, -num_trainable_singular_values:]
        return reduced_U, reduced_S, reduced_V

    def forward(self, x):
        result = self.base_layer(x)
        W = self.U @ self.S.diag() @ self.V.t()
        result += F.linear(x, W)
        return result
```

**(3)** GPT-2 (Medium) (Radford et al., 2019) for natural language generation on the E2E dataset (Wang et al., 2023), and **(4)** LLaMA2-7B (Touvron et al., 2023) for instruction tuning. In each case, along with standard performance metrics for the task, we also report the *performance per parameter*. This quantity indicates how much each trained parameter contributes to the final performance on average. Additionally, we study the effects of: **(1)** using singular vectors derived from original weights versus random matrices, and **(2)** using upper versus lower singular values.

### 4.1 IMAGE CLASSIFICATION

**Models and Datasets.** SiVA is evaluated on the image classification task using the Vision Transformer (ViT) (Dosovitskiy et al., 2021), in both Base and Large variants. The models are pre-trained on the ImageNet-21K dataset (Ridnik et al.), and fine-tuned on OxfordPets (Parkhi et al., 2012)(37 classes), CIFAR10 (10 classes) (Krizhevsky, 2009), EuroSAT (10 classes) (Helber et al., 2019), RE-SISC45 (45 classes) (Cheng et al., 2017), StanfordCars (196 classes) (Krause et al., 2013), FGVC (100 classes) (Maji et al.), and CIFAR100 (100 classes) (Krizhevsky, 2009). We compare SiVA against two established parameter-efficient fine-tuning (PEFT) methods: LoRA (Hu et al., 2022)

and FourierFT (Gao et al., 2024). Additionally, we also show results for training a single Linear Layer on top of the base model (**LP**) and full fine-tuning of the base model (**FF**). We reuse baseline numbers from FourierFT to ensure fairness.

**Implementation Details.** SiVA is evaluated against Full Fine-tuning (FF), Linear Probing (LP, fine-tuning the classification head only), LoRA (Hu et al., 2022), and FourierFT (Gao et al., 2024). For LoRA, FourierFT and SiVA , we fine-tune the query and value matrices in ViT. We report the results for using using $r = 16$ for LoRA, $n = \{3000\}$ for FourierFT, and $perc = 0.95$ for SiVA . Learning rates and weight decay are tuned, with a maximum of 10 training epochs. Hyperparameters are provided in Table 7 in the Appendix.

| Model | Method | # Trainable Parameters | OxfordPets | StanfordCars | CIFAR10 | EuroSAT | FGVC | RESISC45 | CIFAR100 | Avg. |
|---|---|---|---|---|---|---|---|---|---|---|
| ViT-Base | LP | - | $90.28_{\pm0.43}$ | $25.76_{\pm0.28}$ | $96.41_{\pm0.02}$ | $88.72_{\pm0.13}$ | $17.44_{\pm0.43}$ | $74.22_{\pm0.10}$ | $84.28_{\pm0.11}$ | 68.16 |
| | FF | 85.8M | $93.14_{\pm0.40}$ | $79.78_{\pm1.15}$ | $98.92_{\pm0.05}$ | $99.05_{\pm0.09}$ | $54.84_{\pm1.23}$ | $96.13_{\pm0.13}$ | $92.38_{\pm0.13}$ | 87.75 |
| | LoRA (Hu et al., 2022) | 581K | $93.19_{\pm0.36}$ | $45.38_{\pm0.41}$ | $\mathbf{98.78}_{\pm0.05}$ | $\mathbf{98.44}_{\pm0.15}$ | $25.16_{\pm0.16}$ | $\mathbf{92.70}_{\pm0.18}$ | $\mathbf{92.02}_{\pm0.12}$ | 77.95 |
| | FourierFT (Gao et al., 2024) | 72K | $\underline{93.21}_{\pm0.26}$ | $\underline{46.11}_{\pm0.24}$ | $98.58_{\pm0.07}$ | $\underline{98.29}_{\pm0.04}$ | $\underline{27.51}_{\pm0.64}$ | $91.97_{\pm0.31}$ | $\underline{91.20}_{\pm0.14}$ | $\underline{78.12}$ |
| | **SiVA** | 9.7K | $\mathbf{95.16}_{\pm0.53}$ | $\mathbf{59.51}_{\pm0.4}$ | $\underline{98.75}_{\pm0.02}$ | $98.25_{\pm0.04}$ | $\mathbf{47.43}_{\pm1.7}$ | $\underline{92.58}_{\pm0.35}$ | $90.67_{\pm0.10}$ | **83.19** |
| ViT-Large | LP | - | $91.11_{\pm0.30}$ | $37.91_{\pm0.27}$ | $97.78_{\pm0.04}$ | $92.64_{\pm0.08}$ | $24.62_{\pm0.24}$ | $82.02_{\pm0.11}$ | $84.28_{\pm0.11}$ | 72.91 |
| | FF | 303.3M | $94.43_{\pm0.56}$ | $88.90_{\pm0.26}$ | $99.15_{\pm0.05}$ | $99.04_{\pm0.08}$ | $68.25_{\pm1.63}$ | $96.43_{\pm0.07}$ | $93.58_{\pm0.19}$ | 91.4 |
| | LoRA (Hu et al., 2022) | 1.57M | $\underline{94.82}_{\pm0.09}$ | $\underline{73.25}_{\pm0.36}$ | $\underline{99.13}_{\pm0.03}$ | $98.63_{\pm0.07}$ | $\underline{42.32}_{\pm0.98}$ | $\mathbf{94.71}_{\pm0.25}$ | $\mathbf{94.87}_{\pm0.10}$ | $\underline{85.39}$ |
| | FourierFT (Gao et al., 2024) | 144K | $94.46_{\pm0.28}$ | $69.56_{\pm0.30}$ | $99.10_{\pm0.04}$ | $\mathbf{98.65}_{\pm0.09}$ | $39.92_{\pm0.68}$ | $93.86_{\pm0.14}$ | $\underline{93.31}_{\pm0.09}$ | 84.12 |
| | **SiVA** | 30.3K | $\mathbf{95.98}_{\pm0.2}$ | $\mathbf{79.61}_{\pm0.52}$ | $\mathbf{99.16}_{\pm0.07}$ | $\mathbf{98.65}_{\pm0.1}$ | $\mathbf{54.19}_{\pm1.9}$ | $\underline{94.52}_{\pm0.2}$ | $92.39_{\pm0.3}$ | **87.79** |

Table 2: Fine-tuning results with ViT Base and Large models on different image classification datasets. We report the accuracy (%) after 10 epochs. Avg. represents the average accuracy of each method on all datasets. We show the best performance across PEFT methods in **bold**, and underline the next best PEFT method.

**Results.** Table 2 presents the results across seven image classification datasets. All three adapter-based methods significantly outperform Linear Probing. SiVA achieves comparable or better performance with **two orders of magnitude fewer** parameters compared to LoRA and an order of magnitude fewer parameters compared to FourierFT, which is a method that attempts to aggressively reduce number of parameters over LoRA. We also outperform other methods on performance per parameter metric by a significant margin (see Fig 1), achieving state-of-the-art results on all benchmark datasets.

## 4.2 NATURAL LANGUAGE UNDERSTANDING

**Models and Datasets.** SiVA is evaluated on the GLUE benchmark (Wang et al., 2018), covering various natural language understanding tasks such as sentence classification, paraphrase detection, and natural language inference. We fine-tune the RoBERTa-Large model (Liu et al., 2020) for this evaluation. We compare our approach to multiple other parameter-efficient fine-tuning approaches: **Full Fine-tuning (FF)**, where all parameters are updated during fine-tuning, starting from pre-trained weights and biases; **Bitfit** (Zaken et al., 2022), where the biases are tuned while keeping all other parameters fixed; Three variants of **Adapter Tuning** Houlsby et al. (2019a), which introduces two-layer adapters between frozen transformer layers; **LoRA** (Hu et al., 2022), which formulates parameter updates as a product of two low-rank matrices; **DyLoRA** (Valipour et al., 2023) and **AdaLoRA** (Zhang et al., 2023) both of which dynamically optimize the rank of LoRA matrices; **VeRA** (Kopiczko et al., 2024), which employs "scaling vectors" to adapt a pair of frozen random matrices shared between layers for weight updates, and finally, **FourierFT** (Gao et al., 2024), which leverages the Inverse Fourier Transform to learn parameters in the frequency domain and translate them into the weight space.

**Implementation Details.** We train singular values corresponding to the bottom 5% variance of weight matrices (i.e., $perc = 0.95$ across 24 layers). For all six GLUE tasks, we tune the learning rates of the head and SiVA parameters. The fine-tuning setup is similar to Hu et al. (2022), targeting the query and value matrices in each transformer block, with full fine-tuning of the classification head. Hyperparameters are detailed in Table 9 in the Appendix.

**Results.** Table 3 summarizes the results. We present the mean over five random seeds with the best epoch selected. SiVA matches or surpasses baseline methods with significantly fewer trainable parameters, including Full Fine-tuning in some cases, such as for MRPC. We outperform other methods on performance per parameter metric by a large margin (See Fig 3).

| Method | # Trainable Parameters | SST-2 (Acc.) | MRPC (Acc.) | CoLA (MCC) | QNLI (Acc.) | RTE (Acc.) | STS-B (PCC) | Avg. |
|---|---|---|---|---|---|---|---|---|
| FF | 356M | 96.4 | 90.9 | 68 | 94.7 | 86.6 | 92.4 | 88.2 |
| Adpt$^P$ (Pfeiffer et al., 2021) | 3M | $96.1_{\pm0.3}$ | $90.2_{\pm0.7}$ | $68.3_{\pm1.0}$ | $94.8_{\pm0.2}$ | $83.8_{\pm2.9}$ | $92.1_{\pm0.7}$ | 87.6 |
| Adpt$^P$ (Pfeiffer et al., 2021) | 0.8M | $96.6_{\pm0.2}$ | $89.7_{\pm1.2}$ | $67.8_{\pm2.5}$ | $94.8_{\pm0.3}$ | $80.1_{\pm2.9}$ | $91.9_{\pm0.4}$ | 86.8 |
| Adpt$^H$ (Houlsby et al., 2019b) | 6M | $96.2_{\pm0.3}$ | $88.7_{\pm2.9}$ | $66.5_{\pm4.4}$ | $94.7_{\pm0.2}$ | $83.4_{\pm1.1}$ | $91.0_{\pm1.7}$ | 86.8 |
| Adpt$^H$ (Houlsby et al., 2019b) | 0.8M | $96.3_{\pm0.5}$ | $87.7_{\pm1.7}$ | $66.3_{\pm2.0}$ | $94.7_{\pm0.2}$ | $72.9_{\pm2.9}$ | $91.5_{\pm0.5}$ | 84.9 |
| LoRA (Hu et al., 2022) | 0.8M | $\mathbf{96.2}_{\pm0.5}$ | $90.2_{\pm1.0}$ | $\underline{68.2}_{\pm1.9}$ | $\mathbf{94.8}_{\pm0.3}$ | $85.2_{\pm1.1}$ | $\mathbf{92.3}_{\pm0.5}$ | 87.8 |
| FourierFT (Gao et al., 2024) | $\underline{0.048M}$ | $96.0_{\pm0.2}$ | $\underline{90.9}_{\pm0.3}$ | $67.1_{\pm1.4}$ | $\underline{94.4}_{\pm0.4}$ | $\mathbf{87.4}_{\pm1.6}$ | $91.9_{\pm0.4}$ | $\underline{88.0}$ |
| VeRA (Kopiczko et al., 2024) | 0.061M | $\underline{96.1}_{\pm0.1}$ | $\underline{90.9}_{\pm0.7}$ | $68.0_{\pm0.8}$ | $\underline{94.4}_{\pm0.2}$ | $85.9_{\pm0.7}$ | $91.7_{\pm0.8}$ | 87.8 |
| **SiVA** | **0.023M** | $96.2_{\pm0.1}$ | $\mathbf{91.4}_{\pm0.4}$ | $\mathbf{68.4}_{\pm0.9}$ | $94.2_{\pm0.1}$ | $\underline{87.1}_{\pm0.1}$ | $\underline{92.0}_{\pm0.1}$ | **88.22** |

Table 3: Performance of various fine-tuning methods with RoBERTa Large on 6 tasks of the GLUE benchmark. We report the Matthew's correlation coefficient (MCC) for CoLA, Pearson correlation coefficient (PCC) for STS-B and accuracy (Acc.) for all the remaining tasks. We report the mean result of 5 runs with different seeds, each using different random seeds. The best results across PEFT methods for each dataset are shown in **bold**, and the second best results are underlined. Higher is better for all metrics except the number of trainable parameters, where lower is better.

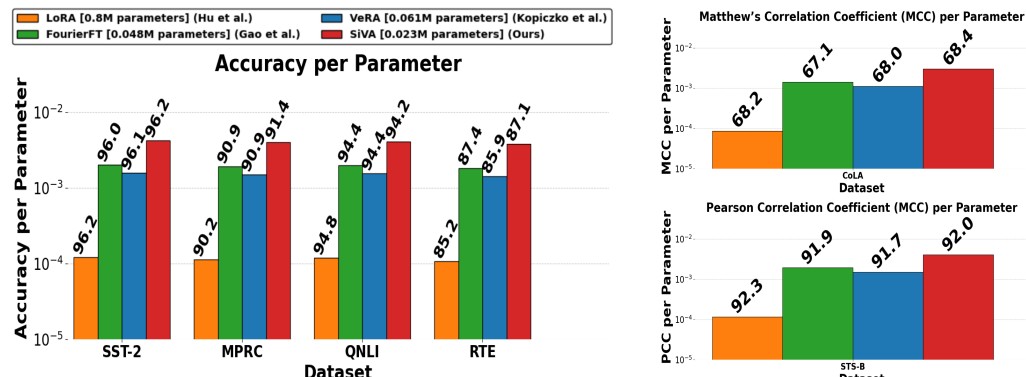

Figure 3: *(Left)* Accuracy per parameter for SST-2, MPRC,QNLI and RTE datasets. Accuracy for each method on each dataset is displayed above the corresponding bar, while number of trainable parameters for each method is provided in legend. *(Right)* Matthew's correlation coefficient (MCC)/ Pearson correlation coefficient (PCC) per parameter for CoLA/STS-B dataset. MCC/PCC for each method is displayed above corresponding bar, while number of trainable parameters for each method is provided in legend. SiVA demonstrates significantly better performance per parameter across the results.

### 4.3 NATURAL LANGUAGE GENERATION

**Models and Datasets.** SiVA is evaluated on the E2E NLG task (Wang et al., 2023), using GPT-2 Medium (354M) model. The E2E dataset includes about 42,000 training samples and 4,600 samples each for validation and testing in the restaurant domain.

**Implementation Details.** We reuse baseline results from previous works, except for LoRA and SiVA , which are fine-tuned using a linear learning rate scheduler over five epochs. The batch size and learning rate are tuned, and the last epoch is selected for evaluation across three runs. Hyperparameters are detailed in Table 8 in the appendix.

**Results.** As shown in Table 4, SiVA achieves comparable or better performance than state-of-the-art methods across all metrics. SiVA does this with the lowest number of parameters, with order of magnitude fewer parameters compared to LoRA.

| Method | # Trainable Parameters | BLEU | NIST | METEOR | ROUGE-L | CIDEr |
|---|---|---|---|---|---|---|
| FT | 354.92M | 68.2 | 8.62 | 46.2 | 71.0 | 2.47 |
| Adpt$^L$ | 0.37M | 66.3 | 8.41 | 45.0 | 69.8 | 2.40 |
| Adpt$^L$ | 11.09M | 68.9 | 8.71 | 46.1 | 71.3 | 2.47 |
| Adpt$^H$ | 11.09M | 67.3 | 8.5 | 46.0 | 70.7 | 2.44 |
| LoRA | 0.35M | 68.9 | 8.76 | $\underline{46.6}$ | $\underline{71.5}$ | **2.53** |
| FourierFT | $\underline{0.048M}$ | 69.1 | **8.82** | **47.0** | 71.8 | $\underline{2.51}$ |
| **SiVA** | **0.044M** | 69.7 | $\underline{8.81}$ | 46.5 | 71.3 | 2.48 |

Table 4: Results with GPT-2 Medium on E2E dataset. For all metrics other than the number of trainable parameters, higher values are better. Best results across PEFT methods are shown in **bold**, and second best results are underlined.

## 4.4 INSTRUCTION TUNING

**Models and Datasets.** Instruction tuning (Ouyang et al., 2022; Wei et al.; Mishra et al., 2022) involves fine-tuning models on paired prompts and responses. We apply VeRA, FourierFT and SiVA to LLaMA2 (Touvron et al., 2023), fine-tuning the LLaMA2-7B variant on the Alpaca dataset (Taori et al., 2023), which contains 51K instruction-following demonstrations. For evaluation, we generate responses to questions from MT-Bench (mtb), with GPT-4 scoring responses on a scale of 10. Since the GPT-4 model has likely changed since the baseline numbers were published, we evalute the generations from baseline methods and SiVA using the model behind the OpenAI API at the time of writing.

**Implementation Details.** For FourierFT, we use $n = 1000$ and use $r = 1024$ for VeRA. For our method, we use $perc = 0.997$. All methods are trained for one epoch. Hyperparameters are provided in Table 10 in the Appendix.

| Model | Method | # Parameters | Score |
|---|---|---|---|
| | VeRA | 327K | **4.41** |
| LLAMA2 7B | FourierFT | 64K | 4.31 |
| | SiVA | **51K** | 4.36 |

Table 5: Scores on MT-Bench Benchmark after tuning on the Alpaca dataset. Scores are provided by GPT-4 as judge.

**Results.** Table 5 shows the results for LLaMA2-7B. We only run on this variant due to compute and budget constraints. SiVA performs on par with other methods, with fewer parameters than either. Practical examples are presented in Appendix A.3.

## 4.5 ANALYSIS

Due to its inherent simplicity, our method comprises few components that can be altered or ablated. We concentrate on two primary aspects informed by our theoretical framework for this analysis: (1) The selection of a random set of singular vectors for constructing $\Delta W$, rather than deriving them from $W$ (referred to as "SiVA -Random"). This approach parallels the methodology of utilizing a random basis as seen in (Gao et al., 2024; Koohpayegani et al., 2024; Kopiczko et al., 2024); and (2) Training the upper singular values instead of the lower ones (designated as "SiVA -Top"). In both scenarios, we maintain the parameter count at each layer consistent with our standard formulation and use the same hyperparameters for training.

We report the results of this analysis in Table 6 for three image classification datasets using ViT-Base. Expectedly, we see a drop performance when we use random singular vectors, and when training the upper singular values.

| Dataset | EuroSAT | FGVC | OxfordPets |
|---|---|---|---|
| SiVA | **98.25**$_{\pm0.04}$ | **47.43**$_{\pm1.7}$ | **95.16**$_{\pm0.53}$ |
| SiVA -Random | 97.98 | 44.31 | 94.29 |
| SiVA -Top | 91.06 | 41.26 | 93.20 |

Table 6: Performance across diff datasets with random bases in $\Delta W$ and by training top singular values

## 5 CONCLUDING REMARKS

In this work, we introduce Singular Value Adaptation (SiVA), a novel, simple and efficient PEFT technique which is grounded in theoretical insights. SiVA increases the effective rank by selectively training only a subset of singular values in the adapter weight matrices, $\Delta W$. The method's efficiency stems from two core principles: (1) *Sparsity*, where instead of updating all singular values of the pre-trained weight matrix $W$, SiVA updates only the $k$-smallest singular values, with $k$ determined adaptively at each layer based on the cumulative sum of singular values and a hyperparameter $perc$. This often leads to $k$ being much smaller than the total number of singular values. (2) *Alignment*, where the singular vectors of the adapter matrix $\Delta W$ are aligned with those of $W$. This alignment reduces the need to train additional parameters, avoiding the necessity of training the singular vectors $U_{\Delta W}$ and $V_{\Delta W}$ as would be required with arbitrary updates. These principles allow SiVA to dramatically reduce the number of trainable parameters while maintaining or exceeding performance on large-scale datasets. Experimental results show SiVA achieves a $2\times$ to $50\times$ reduction in trainable parameters compared to existing PEFT methods like LoRA and FourierFT, while still achieving competitive performance. SiVA's low parameter count makes it especially advantageous for applications requiring frequent model switching or the storage of many fine-tuned models. Reuse of significant parts of the base model makes it ideal for tasks like domain adaptation, continual learning, and multi-task learning, where maintaining a backbone of shared information while adapting to specific downstream tasks is crucial.

As future work, SiVA's parameter efficiency on Transformer architectures can be linked to the architecture's inductive bias (Tarzanagh et al., 2023) and the cross-entropy loss. Additionally, exploring the relationship between SiVA's singular values and the intrinsic rank (Aghajanyan et al., 2021) could help clarify how closely SiVA approaches the optimal number of parameters required for each task.

ETHICAL CONSIDERATIONS AND REPRODUCIBILITY STATEMENT

- **Human Subjects**: Our work does not make use of any human subjects. All datasets used in our work are publicly available and do not involve human subjects.

- **Results**: All results that we report are honestly executed and accurate to the best of our knowledge.

- **Potential Harmful Insights/Methods/Applications**: Our work does not have potential negative impact of harmful applications.

- **Research Integrity**: We have attempted to make best efforts to properly research existing works and ideas in the domain and have compared against contemporary benchmarks fairly.

We have provided the complete pseudocode of our approach in Algorithm 1 for purposes of reproducibility. Additionally, we provide all hyperparameters used for training our models in the Appendix (Tables 7, 9, 10, 8). We shall also release the complete code of our method used to obtain our reported results post acceptance.

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
