# A  APPENDIX

## CONTENTS

## A.1  HYPERPARAMATERS

| ViT-Base | | | | | | | |
|---|---|---|---|---|---|---|---|
| Hyperparameter | OxfordPets | StanfordCars | CIFAR10 | EuroSAT | FGVC | RESISC45 | CIFAR100 |
| Epochs | | | | 10 | | | |
| Optimizer | | | | AdamW | | | |
| *perc* value | | | | 0.95 | | | |
| LR Schedule | | | | Linear | | | |
| Learning Rate (SiVA ) | 1E-1 | 5E-1 | 1E-1 | 1E-1 | 1E-1 | 3E-1 | 1E-1 |
| Learning Rate (Head) | 5E-3 | 5E-3 | 1E-3 | 7E-3 | 1E-1 | 3E-1 | 1E-2 |
| Weight Decay | 1E-4 | 1E-4 | 9E-5 | 1E-3 | 1E-4 | 1E-2 | 1E-2 |
| ViT-Large | | | | | | | |
| Hyperparameter | OxfordPets | StanfordCars | CIFAR10 | EuroSAT | FGVC | RESISC45 | CIFAR100 |
| Epochs | | | | 10 | | | |
| Optimizer | | | | AdamW | | | |
| *perc* value | | | | 0.95 | | | |
| LR Schedule | | | | Linear | | | |
| Learning Rate (SiVA ) | 1E-1 | 1E-1 | 1E-1 | 1E-1 | 1E-1 | 1E-1 | 1E-1 |
| Learning Rate (Head) | 5E-3 | 5E-3 | 1E-2 | 7E-3 | 1E-2 | 3E-2 | 1E-3 |
| Weight Decay | 1E-4 | 1E-4 | 1E-3 | 1E-3 | 1E-3 | 1E-2 | 1E-4 |

Table 7: Hyperparameter setup for image classification for ViT-Base and ViT-Large.

| Hyperparameter | Value |
|---|---|
| Optimizer | AdamW |
| *perc* Value | 0.5 |
| Learning Rate | 0.7 |
| Warmup Steps | 300 |
| Batch Size | 8 |
| Epochs | 5 |
| LR Schedule | Linear |

Table 8: Hyperparameter setup for the E2E benchmark.

| Model | Hyperparameter | SST-2 | MRPC | CoLA | QNLI | RTE | STS-B |
|---|---|---|---|---|---|---|---|
| | Optimizer | | | AdamW | | | |
| | LR Schedule | | | Linear | | | |
| | *prec* Value | | | 0.95 | | | |
| Roberta | Epochs | 30 | 60 | 60 | 30 | 60 | 60 |
| Large | Learning Rate (SiVA ) | 0.009 | 0.015 | 0.01 | 0.004 | 0.01 | 0.008 |
| | Learning Rate (Head) | 1e-4 | 4e-4 | 0.01 | 7e-4 | 6e-5 | 5e-4 |
| | Max Seq. Len | 128 | 512 | 512 | 256 | 512 | 512 |
| | Batch Size | 32 | 32 | 128 | 32 | 32 | 32 |
| | $S_{\Delta W}$ Init | Copy $S_W$ | Copy $S_W$ | Zeros | Zeros | Zeros | Copy $S_W$ |

Table 9: Hyperparameter setup for the GLUE benchmark.

| Hyperparameter | Values |
|---|---|
| Optimizer | AdamW |
| Warmup Ratio | 0.05 |
| Batch Size | 4 |
| Accumulation Steps | 4 |
| Epochs | 1 |
| LR Schedule | Linear |
| Learning Rate | 4e-3 |
| $perc$ Value | 0.997 |

Table 10: Hyperparameter setup for instruction-tuning.

## A.2  THEORETICAL RESULTS

**Lemma 1** (Change in Entropy). *Given a random variable $\chi$ and any two distributions $\mathcal{P}$, $\mathcal{Q}$ over $(\bigcup_{i=1}^{n} \chi_i)$, define $\delta p_i := \mathcal{P}(\chi = \chi_i) - \mathcal{Q}(\chi = \chi_i)$, and assume that such that $|\delta p_i| << 1$, $\forall 1 \leq i \leq n$. Then the change in entropy $\Delta H_{q,p} := H(q) - H(p)$ is given by:*

$$\Delta H_{q,p} = -\sum_{i=1}^{n} \delta p_i (\log p_i)$$

*Proof.* We begin by expanding $H(q)$ as:

$$H(q) = -\sum_{i=1}^{n} q_i \log q_i$$

Substitute $q_i = p_i + \delta p_i$ into this expression:

$$H(q) = -\sum_{i=1}^{n} (p_i + \delta p_i) \log(p_i + \delta p_i)$$

For small perturbations $\delta p_i$, applying a first-order Taylor expansion for $\log(p_i + \delta p_i)$, we have:

$$\log(p_i + \delta p_i) \approx \log p_i + \frac{\delta p_i}{p_i}$$

Thus, $H(q)$ becomes:

$$H(q) \approx -\sum_{i=1}^{n} \left( (p_i + \delta p_i) \left( \log p_i + \frac{\delta p_i}{p_i} \right) \right)$$

Expanding and simplifying the above expression, and ignoring the second-order term of $\delta p_i$,:

$$H(q) \approx -\sum_{i=1}^{n} (p_i \log p_i + \delta p_i \log p_i + \delta p_i)$$

Now, the change in entropy $\delta H$ is:

$$\delta H = H(q) - H(p) = -\sum_{i=1}^{n} (\delta p_i \log p_i + \delta p_i)$$

This simplifies to:

$$\delta H = -\sum_{i=1}^{n} \delta p_i (\log p_i + 1) = \boxed{-\sum_{i=1}^{n} \delta p_i (\log p_i)},$$

where the latter follows since $\sum_{i=1}^{n} \delta p_i = \sum_{i=1}^{n} \mathcal{P}(\chi = \chi_i) - \sum_{i=1}^{n} \mathcal{Q}(\chi = \chi_i) = 1 - 1 = 0$.

$\square$

**Theorem 1** (Optimal Modification of Singular Values to Maximize Entropy Increase). *Let $A \in \mathbb{R}^{n \times n}$ be a matrix with singular values $\sigma_i^A \geq 0$ for $i = 1, 2, \ldots, n$, and let $S_A = \sum_{i=1}^{n} \sigma_i^A$. Consider perturbations $\delta\sigma_i \in \mathbb{R}$ such that the singular values of $A + B$ become $\sigma_i^{A+B} = \sigma_i^A + \delta\sigma_i$, for an arbitrary matrix $B \in \mathbb{R}^{n \times n}$. Under the constraint $\|B\|_F^2 = \sum_{i=1}^{n}(\delta\sigma_i)^2 \leq C$, the maximum possible change in entropy $\Delta H$ is:*

$$\Delta H_{\max} = \sqrt{C} \cdot \sqrt{\sum_{i=1}^{n} c_i^2}, \tag{3}$$

*where*

$$c_i = -\frac{\left(\log\left(\frac{\sigma_i^A}{S_A}\right) + \mathcal{H}_A\right)}{S_A}. \tag{4}$$

*The optimal adjustments $\delta\sigma_i$ are:*

$$\delta\sigma_i = c_i \cdot \frac{\sqrt{C}}{\sqrt{\sum_{j=1}^{n} c_j^2}}. \tag{5}$$

*Further, the coefficients $c_i$ satisfy:*

- $c_i > 0$ if $p_i < e^{-\mathcal{H}_A}$ *(lower singular values).*

- $c_i < 0$ if $p_i > e^{-\mathcal{H}_A}$ *(larger singular values).*

- $c_i = 0$ if $p_i = e^{-\mathcal{H}_A}$.

*Proof.* **1. Expression for Maximum Entropy Increase**

From the previous theorem, the maximum possible increase in entropy when all singular values can be adjusted is:

$$\Delta H_{\max} = \sqrt{C} \cdot \sqrt{\sum_{i=1}^{n} c_i^2}, \tag{6}$$

where

$$c_i = -\frac{1}{S_A}\left(\log p_i + \mathcal{H}_A\right). \tag{7}$$

The optimal adjustments to the singular values are:

$$\delta\sigma_i = c_i \cdot \frac{\sqrt{C}}{\sqrt{\sum_{j=1}^{n} c_j^2}}. \tag{8}$$

**2. Constraint on the Number of Modifiable Singular Values**

When only $k$ singular values can be modified, let $S \subseteq \{1, 2, \ldots, n\}$ be the set of indices of the singular values to be adjusted, with $|S| = k$. The maximum increase in entropy is then:

$$\Delta H_S = \sqrt{C} \cdot \sqrt{\sum_{i \in S} c_i^2}. \tag{9}$$

Our goal is to choose $S$ to maximize $\Delta H_S$.

**3. Relationship Between $c_i^2$ and $p_i$**

The coefficients $c_i$ depend on $p_i$:

$$c_i = -\frac{1}{S_A}\left(\log p_i + \mathcal{H}_A\right). \tag{10}$$

Consider $c_i^2$:

$$c_i^2 = \left(\frac{1}{S_A}\right)^2 \left(\log p_i + \mathcal{H}_A\right)^2. \tag{11}$$

We analyze how $c_i^2$ varies with $p_i$:

- **As $p_i \to 0^+$ (lower singular values)**:

  - $\log p_i \to -\infty$.

  - $\log p_i + \mathcal{H}_A \to -\infty$ (since $\mathcal{H}_A$ is finite).

  - $c_i \to +\infty$.

  - $c_i^2 \to +\infty$.

- **At $p_i = e^{-\mathcal{H}_A}$**:

  - $\log p_i + \mathcal{H}_A = 0$.

  - $c_i = 0$.

  - $c_i^2 = 0$.

- **As $p_i \to 1^-$ (higher singular values)**:

  - $\log p_i \to 0$.

  - $\log p_i + \mathcal{H}_A \to \mathcal{H}_A$ (positive since $H_A > 0$).

  - $c_i = -\frac{\mathcal{H}_A}{S_A} < 0$.

  - $c_i^2 = \left( \frac{\mathcal{H}_A}{S_A} \right)^2$ (finite and relatively small).

**4. Optimal Selection of Singular Values**

To maximize $\Delta H_S$, we need to maximize $\sum_{i \in S} c_i^2$. From the analysis above, $c_i^2$ is largest when $p_i$ is smallest. Therefore, the optimal strategy is to select the $k$ singular values with the smallest $p_i$, i.e., the lowest $k$ singular values.

$\square$

**Theorem 2** (Sparse, Optimal Modification of Singular Values to Maximize Entropy Increase). *Let $A, B \in \mathbb{R}^{n \times n}$ be two matrices, with $A$ being fixed. Suppose we are allowed to perturb at most $k$ singular values of $A + B$ (i.e., at most $k$ of the $\delta\sigma_i$ are non-zero), using a matrix $B \in \mathbb{R}^{n \times n}$ under the constraint $\|B\|_F^2 = \sum_{j=1}^k (\sigma_j^B)^2 \leq C$. To maximize the increase in entropy $\Delta H = \mathcal{H}_{A+B} - \mathcal{H}_A$, it is optimal to modify the $k$-smallest singular values of $A$.*

*Proof.* **1. Expression for Maximum Entropy Increase**

From the previous theorem, the maximum possible increase in entropy when all singular values can be adjusted is:

$$\Delta H_{\max} = \sqrt{C} \cdot \sqrt{\sum_{i=1}^n c_i^2}, \tag{12}$$

where

$$c_i = -\frac{1}{S_A} \left( \log p_i + \mathcal{H}_A \right). \tag{13}$$

The optimal adjustments to the singular values are:

$$\delta\sigma_i = c_i \cdot \frac{\sqrt{C}}{\sqrt{\sum_{j=1}^n c_j^2}}. \tag{14}$$

**2. Constraint on the Number of Modifiable Singular Values**

When only $k$ singular values can be modified, let $S \subseteq \{1, 2, \ldots, n\}$ be the set of indices of the singular values to be adjusted, with $|S| = k$. The maximum increase in entropy is then:

$$\Delta H_S = \sqrt{C} \cdot \sqrt{\sum_{i \in S} c_i^2}. \tag{15}$$

Our goal is to choose $S$ to maximize $\Delta H_S$.

**3. Relationship Between $c_i^2$ and $p_i$**

The coefficients $c_i$ depend on $p_i$:

$$c_i = -\frac{1}{S_A} \left( \log p_i + \mathcal{H}_A \right). \tag{16}$$

Consider $c_i^2$:

$$c_i^2 = \left( \frac{1}{S_A} \right)^2 \left( \log p_i + \mathcal{H}_A \right)^2. \tag{17}$$

We analyze how $c_i^2$ varies with $p_i$:

- **As $p_i \to 0^+$ (lower singular values)**:

  - $\log p_i \to -\infty$.

  - $\log p_i + \mathcal{H}_A \to -\infty$ (since $\mathcal{H}_A$ is finite).

  - $c_i \to +\infty$.

  - $c_i^2 \to +\infty$.

- **At $p_i = e^{-\mathcal{H}_A}$**:

  - $\log p_i + \mathcal{H}_A = 0$.

  - $c_i = 0$.

  - $c_i^2 = 0$.

- **As $p_i \to 1^-$ (higher singular values)**:

  - $\log p_i \to 0$.

  - $\log p_i + \mathcal{H}_A \to \mathcal{H}_A$ (positive since $\mathcal{H}_A > 0$).

  - $c_i = -\frac{\mathcal{H}_A}{S_A} < 0$.

  - $c_i^2 = \left( \frac{\mathcal{H}_A}{S_A} \right)^2$ (finite and relatively small).

**4. Optimal Selection of Singular Values**

To maximize $\Delta H_S$, we need to maximize $\sum_{i \in S} c_i^2$. From the analysis above, $c_i^2$ is largest when $p_i$ is smallest. Therefore, the optimal strategy is to select the $k$ singular values with the smallest $p_i$, i.e., the lowest $k$ singular values.

$\square$

**Theorem 3** (Alignment of Singular Vectors w.r.t Pretrained Weights). *Let $A \in \mathbb{R}^{n \times n}$ be a matrix with singular values $\sigma_i^A$ arranged in descending order ($\sigma_1^A \geq \sigma_2^A \geq \cdots \geq \sigma_n^A \geq 0$) and corresponding left and right singular vectors $\mathbf{u}_i^A$ and $\mathbf{v}_i^A$. Let $B \in \mathbb{R}^{n \times n}$ be a fixed matrix with exactly $k$ nonzero singular values $\sigma_j^B$ (with $\sigma_1^B \geq \sigma_2^B \geq \cdots \geq \sigma_k^B > 0$) and corresponding singular vectors $\mathbf{u}_j^B$ and $\mathbf{v}_j^B$.*

*Under the constraint $\|B\|_F^2 = \sum_{j=1}^{k}(\sigma_j^B)^2 = C$, the maximum increase in entropy $\Delta H = \mathcal{H}_{A+B} - \mathcal{H}_A$ is achieved when the following happen, in order:*

1. *First, the singular vectors of $B$ corresponding to its largest singular value are aligned with the singular vectors of $A$ corresponding to its smallest singular value; specifically, $\mathbf{u}_j^B = \mathbf{u}_{n-j+1}^A$ and $\mathbf{v}_j^B = \mathbf{v}_{n-j+1}^A$ for $j = 1, 2, \ldots, k$.*

2. *Since the largest singular value of $B$ is now aligned, $\Delta H$ is further maximized by aligning the next largest singular value of $B$ with the next smallest singular vector of $A$, and so on recursively, for all $k$ singular values of $B$.*

*Therefore, the largest increase in entropy is achieved by aligning the singular values of $B$ in decreasing order with the singular vectors of $A$ in increasing order of their indices.*

*Proof.* We will prove that the optimal way to maximize the increase in entropy $\Delta H$ is by aligning the largest singular values of $B$ with the smallest singular vectors of $A$ as described.

**1. Expression for the Change in Entropy**

The change in entropy $\Delta H$ can be expressed as:

$$\Delta H = \sum_{i=1}^{n} c_i \delta \sigma_i, \tag{18}$$

where

$$c_i = -\frac{1}{S_A}(\log p_i + \mathcal{H}_A), \tag{19}$$

with $p_i = \frac{\sigma_i^A}{S_A}$ and $S_A = \sum_{i=1}^{n} \sigma_i^A$. The coefficients $c_i$ are positive for lower $i$ (smaller singular values of $A$) and negative for higher $i$ (larger singular values of $A$).

**2. First-Order Perturbation of Singular Values**

Under first-order perturbation theory, the change in the singular values $\delta \sigma_i$ due to $B$ is given by:

$$\delta \sigma_i = \mathbf{u}_i^{A^\top} B \mathbf{v}_i^A. \tag{20}$$

Substituting the singular value decomposition (SVD) of $B$:

$$B = \sum_{j=1}^{k} \sigma_j^B \mathbf{u}_j^B \mathbf{v}_j^{B^\top}, \tag{21}$$

we get:

$$\delta \sigma_i = \sum_{j=1}^{k} \sigma_j^B (\mathbf{u}_i^{A^\top} \mathbf{u}_j^B)(\mathbf{v}_j^{B^\top} \mathbf{v}_i^A). \tag{22}$$

Our objective is to choose the alignment of $\mathbf{u}_j^B$ and $\mathbf{v}_j^B$ to maximize $\Delta H$.

**3. Maximizing $\Delta H$ by Aligning Singular Vectors**

**Step 1: Aligning the Largest Singular Value of $B$ with the Smallest Singular Vector of $A$**

To maximize $\Delta H$, we need to maximize $c_i \delta \sigma_i$ for the terms where $c_i$ is positive and largest. Since $c_i$ is largest for the smallest $\sigma_i^A$ (i.e., for $i = n$), we should maximize $\delta \sigma_n$.

This is achieved by aligning the largest singular value $\sigma_1^B$ with the smallest singular vector $\mathbf{u}_n^A$ and $\mathbf{v}_n^A$, i.e., set:

$$\mathbf{u}_1^B = \mathbf{u}_n^A, \quad \mathbf{v}_1^B = \mathbf{v}_n^A. \tag{23}$$

Then,

$$\delta \sigma_n = \sigma_1^B (\mathbf{u}_n^{A^\top} \mathbf{u}_1^B)(\mathbf{v}_1^{B^\top} \mathbf{v}_n^A) = \sigma_1^B, \tag{24}$$

since $\mathbf{u}_n^{A^\top} \mathbf{u}_n^A = 1$ and $\mathbf{v}_n^{A^\top} \mathbf{v}_n^A = 1$.

For $i \neq n$, $\delta\sigma_i$ receives no contribution from $\sigma_1^B$ because the singular vectors are orthogonal:

$$\delta\sigma_i = \sigma_1^B (\mathbf{u}_i^{A^\top} \mathbf{u}_1^B)(\mathbf{v}_1^{B^\top} \mathbf{v}_i^A) = 0. \tag{25}$$

**Step 2: Aligning the Next Largest Singular Values of $B$**

Next, we align the second largest singular value $\sigma_2^B$ with the next smallest singular vector $\mathbf{u}_{n-1}^A$ and $\mathbf{v}_{n-1}^A$:

$$\mathbf{u}_2^B = \mathbf{u}_{n-1}^A, \quad \mathbf{v}_2^B = \mathbf{v}_{n-1}^A. \tag{26}$$

Then,

$$\delta\sigma_{n-1} = \sigma_2^B, \tag{27}$$

and $\delta\sigma_i^{(2)} = 0$ for $i \neq n-1$.

**Continuing the Process for All $k$ Singular Values**

We continue this process for all $k$ singular values of $B$, aligning each $\sigma_j^B$ with $\sigma_{n-j+1}^A$ by setting:

$$\mathbf{u}_j^B = \mathbf{u}_{n-j+1}^A, \quad \mathbf{v}_j^B = \mathbf{v}_{n-j+1}^A, \quad \text{for } j = 1, 2, \ldots, k. \tag{28}$$

Thus,

$$\delta\sigma_{n-j+1} = \sigma_j^B, \tag{29}$$

and $\delta\sigma_i^{(j)} = 0$ for $i \neq n-j+1$.

**4. Maximizing $\Delta H$**

With this alignment, the change in entropy becomes:

$$\Delta H = \sum_{i=1}^{n} c_i \delta\sigma_i = \sum_{j=1}^{k} c_{n-j+1}\sigma_j^B. \tag{30}$$

Note that $c_{n-j+1}$ and $\sigma_j^B$ form a list arranged in decreasing order. By Rearrangement Inequality, the sum of these products is maximized when the corresponding elements of the lists are paired together.

$\square$

**Theorem 4** (**SiVA** Style Solutions Lie in Set of Minimizers for Linear Regression). *Let $A \in \mathbb{R}^{m \times n}$ be a full-rank matrix with singular value decomposition $A = U_A \Sigma_A V_A^\top$, where $U_A \in \mathbb{R}^{m \times n}$ and $V_A \in \mathbb{R}^{n \times n}$ are orthogonal matrices, and $\Sigma_A \in \mathbb{R}^{n \times n}$ is a diagonal matrix with positive entries $\sigma_{A1}, \sigma_{A2}, \ldots, \sigma_{An}$. Let $x \in \mathbb{R}^n$ and $y \in \mathbb{R}^m$ be given vectors. Define $B = U_A S V_A^\top$, where $S \in \mathbb{R}^{n \times n}$ is a diagonal matrix with entries $\sigma_1, \sigma_2, \ldots, \sigma_n$. Consider the mean squared error loss function*

$$L(\sigma_1, \ldots, \sigma_n) = \|(A + B)x - y\|^2.$$

*Then, $L(\sigma_1, \ldots, \sigma_n)$ is minimized by choosing*

$$\sigma_i = \frac{u_i^\top y}{v_i^\top x} - \sigma_{Ai},$$

*for each $i$ such that $v_i^\top x \neq 0$, where $u_i$ and $v_i$ are the $i$-th columns of $U_A$ and $V_A$, respectively.*

In the proof presented below, we present the pertinent calculations showing the existence of an optimal solution for the **SiVA** form. The formal framework to rigorously prove this has been outlined in the main paper, via a proof by contradiction argument.

*Proof.* Our goal is to find the values of $\sigma_i$ that minimize the loss function

$$L(\sigma_1, \ldots, \sigma_n) = \|(A + B)x - y\|^2.$$

Since $A = U_A \Sigma_A V_A^\top$ and $B = U_A S V_A^\top$, we have

$$A + B = U_A(\Sigma_A + S)V_A^\top.$$

Therefore,

$$(A + B)x = U_A(\Sigma_A + S)V_A^\top x.$$

Let us define

$$z = V_A^\top x \quad \text{and} \quad y' = U_A^\top y.$$

Because $U_A$ and $V_A$ are orthogonal matrices ($U_A^\top U_A = I$ and $V_A^\top V_A = I$), the loss function becomes

$$L(\sigma_1, \ldots, \sigma_n) = \|U_A(\Sigma_A + S)z - y\|^2 = \left\|(\Sigma_A + S)z - U_A^\top y\right\|^2 = \|(\Sigma_A + S)z - y'\|^2.$$

Since $\Sigma_A + S$ is a diagonal matrix with diagonal entries $\tilde{\sigma}_i = \sigma_{Ai} + \sigma_i$, we can express the loss function as a sum of squares:

$$L(\sigma_1, \ldots, \sigma_n) = \sum_{i=1}^{n} (\tilde{\sigma}_i z_i - y'_i)^2,$$

where $z_i = v_i^\top x$ and $y'_i = u_i^\top y$.

To minimize $L$ with respect to $\sigma_i$, we consider each term individually:

$$L_i(\sigma_i) = (\tilde{\sigma}_i z_i - y'_i)^2.$$

For each $i$ such that $z_i \neq 0$, we set the derivative of $L_i$ with respect to $\sigma_i$ to zero:

$$\frac{\partial L_i}{\partial \sigma_i} = 2(\tilde{\sigma}_i z_i - y'_i)z_i = 0.$$

Solving for $\sigma_i$, we obtain:

$$\tilde{\sigma}_i z_i = y'_i \implies \sigma_{Ai} + \sigma_i = \frac{y'_i}{z_i} \implies \sigma_i = \frac{y'_i}{z_i} - \sigma_{Ai}.$$

For $i$ such that $z_i = 0$ (i.e., $v_i^\top x = 0$), the term $L_i(\sigma_i)$ becomes independent of $\sigma_i$:

$$L_i(\sigma_i) = (\tilde{\sigma}_i \cdot 0 - y'_i)^2 = (-y'_i)^2.$$

In this case, any value of $\sigma_i$ minimizes $L_i$, as $L_i$ does not depend on $\sigma_i$.

Therefore, the loss function $L$ is minimized by choosing

$$\sigma_i = \frac{u_i^\top y}{v_i^\top x} - \sigma_{Ai},$$

for each $i$ where $v_i^\top x \neq 0$. $\qquad\square$

## A.3 EXAMPLE OF INSTRUCTION-TUNED MODEL GENERATION

We provide a practical example generated by the LLaMA2-7B model that is fine-tuned with FourierFT and SiVA . The example includes a two-turn question from MT-Bench, the answer and the review assigned by GPT-4.

**Question:**

**Turn 1:** Draft a professional email seeking your supervisor's feedback on the 'Quarterly Financial Report' you prepared. Ask specifically about the data analysis, presentation style, and the clarity of conclusions drawn. Keep the email short and to the point.

**Turn 2**: Take a moment to evaluate and critique your own response.

---

**FourierFT - Turn 1**

Dear [Supervisor],
I am writing to request your feedback on the Quarterly Financial Report I prepared for the company. I would appreciate your comments on the data analysis, presentation style, and the clarity of conclusions drawn.
The report can be found attached. I have analyzed the company's financial data over the past quarter and have drawn the following conclusions:
The company's revenue increased by 5% over the previous quarter, with significant growth in the consumer products division.
The company's operating expenses decreased by 2% over the previous quarter, resulting in an increase in net income of 3%.
The company's cash flow increased by 10% over the previous quarter, indicating a healthy financial position.
I have used charts and graphs to visualize the data and make it more accessible to readers. I have also provided detailed explanations of the data analysis in the report.
I would appreciate your feedback on the report's presentation and clarity of conclusions. Your comments will help me improve my work and contribute to the company's financial success.

---

**FourierFT - Turn 2**

My response is well-structured and professional. The email is short and to the point, with a clear request for feedback on the report. I have provided a summary of the data analysis, presentation style, and the clarity of conclusions drawn. The report is attached to the email for easy reference.
Overall, I believe my response is effective in seeking feedback on the report and communicating my professionalism and attention to detail.

---

**FourierFT - GPT4 Evaluation (Rating: 9)**

The assistant's response is highly relevant, accurate, and detailed. It follows the professional tone and format of an email, and it addresses the user's request to seek feedback on the data analysis, presentation style, and clarity of conclusions drawn in the 'Quarterly Financial Report'. The assistant also provides a brief summary of the report's content, which is a good practice in professional communication. However, the assistant could have been more creative by adding a line about being open to improvements or eager to learn from the supervisor's feedback. Overall, the response is very helpful and to the point.

**SiVA - Turn 1**

Dear [Supervisor],
I am writing to request your feedback on the Quarterly Financial Report I prepared for the [company name] Board of Directors. I have attached the report for your review.
I have analyzed the financial data for the past quarter, and have made some observations and recommendations based on the findings. I have also included a summary of the report in the body of the email.
I am always eager to learn and grow, and would appreciate your feedback on my analysis and presentation style. Please let me know if there are any questions or concerns you have about the report.
Thank you for your time and consideration, and I look forward to your feedback.

**SiVA - Turn 2**

I believe that the email was clear and concise, and it included all the relevant information. The summary in the body of the email was well-written and succinct. I also think that the tone of the email was respectful and professional.

**SiVA - GPT4 Evaluation (Rating: 10)**

I believe that the email was clear and concise, and it included all the relevant information. The summary in the body of the email was well-written and succinct. I also think that the tone of the email was respectful and professional.