# OpenReview forum: "Singular Value Adaptation for Parameter-Efficient Fine Tuning"
_ICLR.cc/2025/Conference — ICLR 2025 Conference Withdrawn Submission_

### Official Review · Reviewer_mhpm · 2024-10-17

**Soundness:** 2
**Presentation:** 1
**Contribution:** 2
**Rating:** 3
**Confidence:** 5

**Summary:**

This paper proposes a new PEFT method SiVT that effectively saves parameter budgets while achieving comparable performances compared with other methods.

**Strengths:**

1. Interesting observation.
2. Interesting theory.
3. Clear presentation on Method section.
4. Extensive experiments.

**Weaknesses:**

I would like to raise my score if the authors could address my concerns as below.

Weaknesses 2, 3, 4 8 and 9 are directly related to my rating, and I kindly ask the authors to prioritize addressing these. The other weaknesses are more about improving the overall quality of this manuscript.

Weakness

1. Line 45-46, “no studies on how different formulations". Please refer to [1], Sec.4.1.
2. Line 203: “Under the constraint… ≤C”, why are the singular values of  B \delta \sigma? Even according to singular value perturbation theory, the singular values of the resulting matrix from the sum of two matrices are not the sum of the singular values of the individual matrices. (BTW, the property of singular value perturbation is quite different to that of a constant.)
3. I’m quite confused how the authors derived the relationship between the maximum value and the optimal solution, in Eq (1)-(2). I also reviewed the equations (6)-(8) in the appendix, but I believe the authors did not provide a proof. Even when using the Cauchy-Schwarz inequality, I am unable to reach the same conclusion as the authors and I have carefully checked my reasoning process. For now, I just assume the authors’ proof is correct, but I hope to receive clarification during the rebuttal.
4. Line 161: “the effective rank of the combined … the downstream task”. Why? I do not understand why the effective rank is associated with the performance.  I also tested the performance of LoRA myself using various random seed on datasets CoLA and MRPC and calculate the corresponding effective ranks for all layers of DeBERTaV3-base, but found no obvious relations. Moreover, regarding effective rank, I believe that if the weight matrix of the model has a higher effective rank, it indicates that the matrix retains more degrees of freedom, allowing it to capture a greater variety of features and patterns in high-dimensional space. However, higher is not always better—a high rank could imply that the model is overly complex, leading to overfitting (i.e., the model becomes too well-tuned to the training data, resulting in poor generalization).
5. How is the effective rank of SiVT on the authors’ experiments compared with other methods over other models? A corresponding validation lacks.
6. I acknowledge that SiVT requires only few parameters for training, but I wonder is it possible to conduct an experiment under a similar parameter budget?  I can also understand if it’s difficult for the authors, as some methods with the smallest parameter counts (e.g., LoRA with  r=1 ) may already have a larger parameter size compared to SiVT.
7. The biggest issue with the paper is that it is overly complex, while the proposed method is not as impressive as it seems. In fact, putting aside whether the theory is correct, if the authors agree with my perspective on SiVT mentioned in Weakness 8, then fine-tuning the bottom singular values or components of W is a natural process. Simply, the bottom singular values of W correspond to less important parts from the pre-training. During the learning process of downstream tasks, these parts are naturally adjusted without altering the top components, as the success of fine-tuning relies on the foundation laid by pre-training.
8. In a nutshell, what SVT actually does is just to focus on learning the variation in the bottom singular values of W.  Thus I’d like to see a comparison between SiVT with SVDiff [2], which adjusts all the singular values.
9. I have one very critical question: in fact, even when training n singular values, the parameter count is still very small compared to methods like LoRA and others (referencing SVDiff [2] parameter size). According to the authors’ claim, in this case, its effective rank would be larger than just adjusting k singular values, so the (SVDiff) performance should be better. In this case, is it really necessary to select only k singular values?

[1] 2024, See Further for Parameter Efficient Fine-tuning by Standing on the Shoulders of Decomposition

[2] 2023, SVDiff: Compact Parameter Space for Diffusion Fine-Tuning

Typos:

1. ”effective rank”, the left quotation mark.

**Questions:**

Please refer to weaknesses.

---

### Official Review · Reviewer_Cciw · 2024-10-23

**Soundness:** 3
**Presentation:** 3
**Contribution:** 2
**Rating:** 6
**Confidence:** 4

**Summary:**

This paper proposes SiVA, a novel PEFT method grounded in theoretical analysis of how weight updates affect a matrix's effective rank. The authors establish a theoretical framework showing that modifying smaller singular values of weight matrices, while keeping their singular vectors aligned with the original ones, can efficiently increase the effective rank and improve model performance. The method achieves comparable or better performance than existing PEFT approaches while using significantly fewer parameters.

**Strengths:**

1. This paper establishes a solid theoretical systematically analyzing how singular value modifications affect effective rank, providing mathematical justification for their approach rather than relying on heuristics like many existing PEFT methods.
2. The proposed method achieves remarkable parameter efficiency through theoretically motivated designs, while maintaining or improving performance across tasks.
3. Despite its theoretical sophistication, the method is elegantly simple to implement, requiring only one initial SVD decomposition and straightforward gradient descent on a small number of diagonal elements.

**Weaknesses:**

1.	The fundamental premise linking effective rank to model performance lacks rigorous justification. The correlation between effective rank and downstream performance is demonstrated on a single dataset and only for Query and Value matrices in Table 1, which is insufficient to establish it as a general principle. It is unclear why increasing effective rank would necessarily lead to better task adaptation, and whether this rule applies to weight matrices other than query and value matrices.
2.	In Figure 1 (left), the reported rank changes are shown as small decimal values (e.g., +0.05, +0.90), which is puzzling given that matrix rank should be an integer representing the number of non-zero singular values. The authors should clearly explain their methodology for computing these rank values.
3.	The authors should clarify which transformer layers were analyzed in Figure 1 (left), as parameter ranks can vary significantly between shallow and deep layers, making the layer selection crucial for validating their hypothesis about effective rank.
4.	The experimental results in Table 2 raise serious concerns about fairness in method comparison. The reported accuracies for LoRA on StanfordCars (45.38%) and FGVC (25.16%) are surprisingly low, as my reproduction attempts easily achieve 80%+ and 50%+ respectively. The authors need to provide hyperparameters that reproduce these baselines. Additionally, the authors' decision to compare SiVA's 5-epoch results with 10-epoch baselines (LP, FF, LoRA, FourierFT) requires justification.
5.	The evaluation on GLUE benchmark notably omits QQP and MNLI tasks, which are among the largest and most challenging tasks in this benchmark. Given that these tasks typically require more sophisticated semantic understanding and have substantially larger training sets, their inclusion would provide more convincing evidence of the method's scalability and effectiveness on complex language understanding problems.

**Questions:**

Kindly refer to the weaknesses.

---

### Official Review · Reviewer_PcYT · 2024-10-25

**Soundness:** 2
**Presentation:** 2
**Contribution:** 2
**Rating:** 3
**Confidence:** 4

**Summary:**

This paper aims to improve the effectiveness of LoRA fine-tuning.
The authors assume that a network's performance is related to the effective rank of the corresponding weight matrix and propose increasing the effective rank in LoRA fine-tuning.
Experiments on image classification and NLP are conducted to evaluate their proposed method.

**Strengths:**

- the idea of using effective rank is interesting to me. However, the assumption (performance depends on the effective rank) is not convincing.
- the proposed method is derived from the theoretical analysis in Section 3.2, which studies the relation between effective rank and the updated matrix

**Weaknesses:**

- This paper is based on a new assumption (L146): Our work is based on the hypothesis that **the performance of the adapted model is proportional to the increase in effective rank caused by adding the learned matrix ∆W to the pre-trained weight matrix W.**. This is a very strong assumption, in my opinion, it needs a careful justification before using it to motivate the new algorithm. However, the authors spend little content on it (only Table 1, on a very simple dataset, Stanford cars). I would like to recommend the authors establish this assumption theoretically, otherwise, this paper is a clear rejection.
- the writing is not professional, for example, in the Introduction section, many citations are missing, e.g., L31, L35, L36, L41...
- many LoRA variants have been proposed but not compared with in experiments, e.g., DoRA, AdaLoRA, Delta-LoRA, GaLore
- the GLUE benchmark is sensitive to random seeds, thus, the tiny improvements over baselines are not convincing

**Questions:**

see above

---

### Official Review · Reviewer_RcvT · 2024-11-04

**Soundness:** 3
**Presentation:** 2
**Contribution:** 2
**Rating:** 3
**Confidence:** 3

**Summary:**

This paper proposes a new PEFT method based on SVD called SiVA.
SiVA is built upon *Effective Rank*, a metric that quantifies the uniformity of singular values for a squared matrix.
In specific, the authors propose to increase Effective Rank of weight matrices in a pre-trained model, based on empirical findings that there is a positive correlation between Effective Rank and model performance.
To this end, SiVA only updates the minor singular values of pre-trained matrices, thereby achieving better parameter efficiency.

**Strengths:**

1. The problem studied in this paper is important.
2. The proposed method achieved better parameter efficiency on a variety of tasks.

**Weaknesses:**

After reading through the paper, I am still unconvinced about the theoretical analysis in this paper, which was claimed as a main contribution.

1. The core motivation of SiVA lies in increasing Effective Rank. More broadly, the observation that "model parameters are nearly full-rank" (Line 48), but this is contrastive to the common belief that pre-trained model parameters are often low-rank [1]. Therefore, this observation should be elaborated more, and additional experiment results for support is expected. Currently the authors provided some evidence of how Effective Rank correlates with model performance, but they are actually from different PEFT methods. As a result, this observation may be a spurious correlation. Instead, the authors should use one fixed PEFT method to study how Effective Rank correlates the model performance.

2. Given that the theoretical analysis is solely based on the assumption that "*Effective Rank implies better model performance*", the claims that "*SiVA is more principled than existing method*"/ "*SiVA offers an optimal balance between performance and efficiency*" in Abstract and Introduction sound too strong to me: Effective Rank is essentially a measure similar to condition number (but is more comprehensive, as it considers all singular values instead of just the max-min range). But as mentioned above, the connection between this "rank"  and "performance" is not well-established. Moreover, there are alternative ways to measure this "rank", such as condition number. But it is untouched if increasing Effective Rank is indeed more effective than minimizing condition number --- That is to say, SiVA as a method that maximizes Effective Rank is "optimal", but it doesn't say much  about SiVA being an optimal choice for LLM fine-tuning.

Because of the above two points, I am not convinced that SiVA is more principled than other LoRA variants. Its higher parameter efficiency is established and can be highlighted more.

In addition, I feel that experiments can be enhanced from two aspects:

3. While it is amenable that the authors studied the proposed method on both vision and language models, large language models (LLMs) were rarely experimented. More base models should be included. For example, in Table 4, why did the authors use GPT-2 given that diverse LLMs are available?

4. As mentioned in point 1, more ablation studies on SiVA and Effective Rank should be conducted. For example, how different choices of *perc* would affect SiVA performance, and how its Effective Rank is associated with model performance in more diverse scenarios should be presented.

[1] Intrinsic dimensionality explains the effectiveness of language model fine-tuning, 2020.

**Questions:**

1. Effective Rank measures how uniformly the singular values distribute. Besides increasing the minor singular values, another possible solution is to reduce the major singular values. How did your theoretical analysis handle this?


2. In Line 76, the authors said SiVA "constrains the number of trainable parameters", but in theoretical analysis, the constraint is in fact on B's Frobenius norm. Can you clarify more on how do these two match?


3. What does "Value" mean in Figure (left)? What are these percentages?


Please see above weakness.

---

### Note · Authors · 2024-11-25

I have read and agree with the venue's withdrawal policy on behalf of myself and my co-authors.